# The Effect of Vocational Counseling Interventions for Adults with Substance Use Disorders: A Narrative Review

**DOI:** 10.3390/ijerph19084674

**Published:** 2022-04-13

**Authors:** Min Kim, Andrew M. Byrne, Jihye Jeon

**Affiliations:** 1Department of Social Welfare, Incheon National University, Incheon 22012, Korea; kimmin14@bu.edu; 2School of Education, California Polytechnic State University, San Luis Obispo, CA 93407, USA; anbyrne@calpoly.edu

**Keywords:** substance use disorders, effectiveness of vocational counseling interventions, incorporated treatment, employment outcomes

## Abstract

The authors reviewed the research literature evaluating the effectiveness of vocational counseling interventions focused on employment for consumers with substance use disorders. This review included 11 articles related to vocational counseling interventions, which are either incorporated with substance use treatment or not. The results of this review revealed that vocational counseling services have been highly efficacious in resulting in part-time and full-time jobs. The study designs had some limitations, and few studies employed randomized control trials (RCT).

## 1. Introduction 

The National Survey on Drug Use and Health [1] indicated that 19.7 million Americans aged 12 or older had substance use disorder (SUD) in 2017. Researchers indicated that 40% to 50% or more of those with SUDs have co-occurring psychiatric disorders, which cause more complex challenges among those with the disorder and for society [2,3]. People who overused alcohol and used illicit drugs showed common issues (e.g., hospitalization, unemployment, and mental health problems) and a failure to fulfill major role obligations at work, school, or home.

Hogue and colleagues [4] indicated that one major burden for people with SUDs is unemployment or underemployment, which is critically related to economic independence and stable community lives. Sigurdsson and colleagues reported that the employment rate of those with SUDs is 40%, while the U.S. employment rate is about 90% [5]. Moreover, those with co-occurring psychiatric disorders have even lower employment rates at 25%. According to an analysis of treatment admissions data from 1993 to 2006, people with SUDs have one of the lowest employment rates among those with different types of disabilities in the U.S. [6]. Regardless of the characteristics of those with SUDs (e.g., age, education level, gender, and economic status), this issue has required specialized support for over two centuries [7]. The economic cost of substance use issues was estimated at $193 billion in 2014 [8] and $500 billion in 2020 [9] and includes lost productivity, criminal justice costs, and healthcare costs. These costs continue to increase due to service complexity, including family, vocational, medical, and social issues [10].

In order to reduce the severity of these issues, researchers have developed and evaluated multidisciplinary recovery processes [11,12]. As members of multidisciplinary teams, for example, vocational rehabilitation counselors who provide substance abuse/addiction and vocational counseling services help to reduce substance use behaviors and improve employability [13]. With the aid of vocational rehabilitation counselors, those with SUDs can face their issues directly, develop skills, and build support systems for prolonged success and recovery. However, vocational counseling services have not traditionally been focused on in the substance use field, and researchers have placed emphasis on substance use treatments to reduce consumers’ substance use symptoms and behaviors without considering employment [14]. This review evaluates and synthesizes the evidence for vocational counseling interventions’ impact on SUD services and co-occurring disorders. 

## 2. Literature Review

Vocational counseling interventions have limited recognition for their positive effects, such as improving employability and maintaining a healthier lifestyle, on clients undergoing substance use disorder treatment [12]. Specifically, after completing vocational counseling interventions, consumers are more likely to improve both employment and quality of life [15,16,17,18,19,20] or improve quality of life [20,21]. 

McIntosh and colleagues explored the effects of vocational counseling interventions on people with SUDs [12]. They found increases in self-esteem and positive self-image after completing the interventions. During the interventions, consumers put more effort into seeking and engaging in treatment, thereby experiencing reductions in their addiction behaviors as well as improving vocational skills. The authors emphasized that when receiving vocational counseling interventions and substance use treatment, consumers’ recovery was facilitated, and their performance level at work was increased. Moreover, researchers found that vocational counseling interventions encouraged engagement in substance use treatment [22]. That is, vocational counseling interventions would help consumers to visualize a positive future, such as recovery and economic independence [23,24].

Vocational counseling is useful in developing SUD recovery plans [11,25]. The process of vocational counseling begins with job preparation and placement based on the assessment of job interest. Additionally, continuous training and assessment are required to improve employability and a transition from job preparation to entering the workforce. Vocational rehabilitation counselors coordinate with other practitioners to maximize consumers’ employability [13]. Although new models have been suggested (e.g., individualized placement and support and integrated treatments), vocational counseling is still the most widely available in the field for competitive integrated employment [26]. Researchers have reported that vocational counseling interventions are the most highly preferred service after completing substance use treatment [4,19]. Given that employment is a primary goal for addiction and drug treatment, it follows that successful employment is one of the criteria to measure treatment success [27]. Employment has been used as a successful treatment outcome measure in both substance use and co-occurring psychiatric fields [28]. Magura emphasized that employment was used as a success measure in multiple studies that did not provide vocational counseling interventions or other specialized employment services [22]. Moreover, many studies have compared pre- and post-employment status and evaluated consumers’ employment rates [5,12,24]. Specifically, these employment-related criteria include (1) working hours (i.e., per day or per week); (2) income level; (3) length of ongoing employment (i.e., less or more than 90 days); and (4) type of employment (unemployment, part-time, and full-time employment). Researchers have disseminated evidence that supports vocational counseling interventions in substance use [19,20] and co-occurring psychiatric disorder treatment milieus [17,22]. 

Although a few studies have supported the effectiveness of substance use treatment focused on vocational counseling interventions, this line of research can be challenging [25,29] due to reduced opportunities to move laboratory-based practices into the field for practical replication and scaling [11,14]. Researchers have instead focused their energies on trying to optimize substance abuse treatments to reduce substance abuse and psychiatric symptoms [29] and have neglected to study alternatives including vocational counseling [30]. This review compiles research on the effects of vocational counseling interventions on treatment outcomes for consumers with substance use disorders and co-occurring psychiatric disorders.

## 3. Method for Conducting the Evidence-Based Review

To ensure the comprehensive inclusion of studies related to vocational counseling interventions in this review, a search-by-hand strategy was applied. Three authors independently searched electronic articles published between January 2002 and January 2021 (20 years) from the following journals: *Archives of Physical Medicine and Rehabilitation*; *American Journal of Occupational Therapy*; *Clinical Rehabilitation, Disability and Rehabilitation, and Medicine*; and *Rehabilitation*, *Physical Therapy*, *Psychiatric Services*, and *Substance Use and Misuse*; *Rehabilitation Counseling Bulletin*; *Journal of Rehabilitation*; *Rehabilitation Research*, *Policy*, and *Education*. The authors also investigated articles using the following keywords: “employment service(s)”, “job skill(s) and training”, “prevocational intervention”, “vocational counseling”, and “work outcome(s)”. Additionally, terms related to substance use disorders included: “addiction”, “substance abuse”, “SUD(s)”, and “co-occurring psychiatric disorder(s)”.

### Study Selection Criteria

After collecting articles and abstracts, inclusion/exclusion criteria were used to select only those potentially relevant for a more detailed review of the full text. Inclusion criteria were: (1) published or in-press articles written in English, (2) articles exploring the effects of vocational counseling interventions on SUDs and co-occurring psychiatric disorders, (3) studies including adults over 18 years of age, (5) articles published or publishing in peer-reviewed journals, and (6) those whose full texts were available. 

Exclusion criteria were (1) only substance use treatments outside of the scope of vocational counseling practice, (2) only abstracts without full-text available, (3) one-day-only or one-time-only training or interventions, and (4) research subjects who were diagnosed with only mental illness or physical disability. Where it was unclear whether the study met the inclusion/exclusion criteria, e-mails were sent to the original authors to collect clear information (i.e., study method, intervention, participant, and result).

To categorize research studies related to vocational counseling interventions in SUDs and co-occurring psychiatric disorders, the authors used level-of-evidence definitions based on evidence-based medicine standards [31] as a ranking system, but only the first three levels were included: levels I, II, and III. Descriptive studies as defined by level IV and case reports and expert opinions as defined by level V were excluded (Table 1). 

Then, to develop criteria for the classification of vocational counseling interventions, one study [32] was used: (1) comprehensive work programs, (2) employment counseling/education programs, (3) supported employment plus skills training, and (4) comprehensive substance abuse treatment. An initial search of six databases yielded 102 citations and abstracts written in English that discussed vocational counseling interventions for consumers with SUDs or co-occurring psychiatric disorders (Figure 1). Through the first review, 32 duplicate articles were removed, and then 70 studies meeting exclusion criteria were excluded. The articles selected for inclusion were reviewed and analyzed. Therefore, 11 studies were selected for final analysis to explore which vocational counseling interventions are supported by research to improve employment outcomes for consumers with SUDs and co-occurring disorders. Of those, 6 studies were classified as level I, and 5 studies were level III (see Table 2).

## 4. Results

### 4.1. Comprehensive Work Programs

A level I systematic review [32] included updated studies in three categories: (1) vocational problem solving, (2) job-seeker workshops, and (3) supported work through veterans’ industries. The results of comprehensive work programs were higher rates of employment, more total earnings, and longer job duration. However, because of the unclear descriptions (i.e., study period, service plans, and strategies), the interpretation of results and replication were difficult. 

Butler et al. [33] conducted a level I RCT pre-posttest study for consumers with SUDs from six residential and outpatient treatment sites in five states (*n* = 194). This study provided a work-it-out program by administering computerized comprehensive work programs including motivational segments, job-seeking decisions, job getting, job keeping, and relapse prevention to the treatment group. The control group only received a printed package comprising a booklet. By comparing baseline and 6-month follow-up, the experimental group was found to have significantly reduced levels of Addiction Severity Index (ASI) scores (*p* < 0.001). 

In addition, a level III one-group study by Kemp et al. [34] showed good evidence for comprehensive work programs using Helping Offenders Work (HOW). HOW is a manual-driven series involving job and life skill development, job training technicians, and welfare to work (*n* = 245). Overall, 78% completed the programs, and 55% were employed in competitive jobs. The programs showed moderate effectiveness, while efficacy was not determinable because it was a nonrandomized group trial. 

### 4.2. Employment Counseling and Education Programs

Two studies examined the effectiveness of employment counseling and education programs through group comparisons and found evidence of their effectiveness. A level I RCT, which compared three intervention groups without a control group, was conducted by Lidz et al. [35]. Each group received one of three interventions: (1) 167 participants received vocational problem-solving training to promote job holding, (2) 68 received job-seeker workshop training to improve job-seeking and interviewing skills, and (3) a third group of 66 participants received both interventions. Although all groups reported significant improvement in days of work and a decrease in criminality when completing the intervention, no significant differences were found among groups in 6- and 12-month follow-ups. Pre-vocational interventions for consumers with SUDs were effective, but with no control group, the generalizability of the study results was limited (*p* < 0.05).

A level III group intervention was conducted by Kidorf et al. [36]. Study participants with opioid dependence received *motivated stepped care* (MSC) with intensified individual and group counseling (*n* = 254). MSC interventions were focused on (1) reducing drug and alcohol use levels, (2) increasing employability through education, and (3) improving participant motivation for job seeking. When participants achieved stability in alcohol and drug use frequencies, they were stepped up to employment counseling and education toward an employment outcome. When they were employed, consumers received continuous case management by clinicians to reduce work-related challenges. In order to achieve employment, a model from *T**he Job Club Counselor’s Manual: A Behavioral Approach to Vocational Counseling* [41] was applied. After completing interventions, 70% had full-time employment, and 19% had part-time employment, while 4% had obtained volunteer positions. The favorable results could not be generalized due to limitations in the sample design, including the lack of a control group. 

### 4.3. Supported Employment plus Skills Training

Three studies examined the effectiveness of supported employment programs with vocational skills training. A level I RCT study conducted by Staines et al. [37] compared interventions for people with SUDs in a comprehensive employment supports (CES) program with those in non-intensive vocational counseling (*n* = 121). The CES program focused on intensive individual counseling, utilizing the principles of rapid job search, case management, and supported employment. At 6-month follow-up, the study showed that participants in CES were more likely to have entered paid employment (66% vs. 41%, χ2 = 7.24, *p* < 0.01), competitive employment (27% vs. 14%, χ2 = 3.51, *p* < 0.01), and informal employment (56% vs. 27%, χ2 = 10.54, *p* < 0.001) than those in the non-intensive program. 

A level I RCT study conducted by Davis et al. [28] focused on veterans with co-occurring post-traumatic stress disorder (PTSD). The intervention group received individualized placement and support (IPS, *n* = 42), including rapid job search and placement in competitive jobs. The control group received vocational rehabilitation programming (VRP, *n* = 43) utilizing prevocational testing and evaluation. At 12-month follow-up, the results indicated that 76% of the IPS participants gained competitive employment compared with 28% of the VRP participants (*p* < 0.05). Additionally, the authors indicated that participants in IPS were likely to work longer weeks and earn more income than those receiving VRP.

A level III one-group comparison study conducted by Kerrigan et al. [38] examined the effects of work therapy, including job readiness training, rapid job search, job placement, and drug-free supported housing (*n* = 80). Veterans with SUDs received group work therapy twice per week for four weeks. The results indicated that 54% had competitive jobs after completing the treatment, and 32% had maintained a competitive job at 3-month follow-up. A limitation was that there was no comparison group.

### 4.4. Incorporated Substance Use Treatment

Evidence regarding the effectiveness of substance use services focused on vocational counseling interventions is limited. Coviello et al. [39] conducted a level I RCT for a two-group comparison. The treatment group received vocational problem-solving training and methadone treatment (*n* = 62), and the control group received only methadone treatment (*n* = 47). Although it seemed that the treatment group’s motivation level and the number of job-seeking activities to enter the workforce had increased after completing treatment, no significant differences were found. However, logistic regression analysis showed that levels of motivation (*p* < 0.05, *p* < 0.01) and the number of job-seeking activities during treatment (*p* < 0.05, NS) were significant predictors for obtaining part-time (1–14 days/month) and full-time jobs (15 to 30 days/month) at the 6-month follow up, respectively.

A study by McLellan et al. [40] found positive results in a level III study of *CasaWorks for Families* (CWF) interventions. The CWF interventions provided substance use, employment, and domestic violence programs to women who experienced violence (*n* = 529). With measurements taken at baseline and 6-month and 12-month intervals, significant improvements were found, as more than 46% had abstained from alcohol and other drugs (*p* < 0.001), while 30% were employed at least part-time (*p* < 0.001). Additionally, the participants had significant increases in total wages (*p* < 0.001) and decreases in medical and psychiatric issue levels (*p* < 0.01). However, the one-group structure of this study reduced its generalizability. 

Lastly, a level III one-group study comparing pre- and posttests after completing substance use treatment focused on job readiness and vocational counseling interventions from *Project Working Recovery* (PWR) with 313 participants was conducted by Kim [11]. At 7-month follow-up, the results showed that participants who received coordinated treatment programs showed significant decreases in alcohol and/or drug use and psychiatric symptom levels as measured by the Addiction Severity Index 5 (ASI-5) and increases in employment rates, either part-time or full-time. At baseline, of the 108 participants who completed services, 8 were employed part-time; at the 7-month interval, 10 and 16 reported part-time and full-time employment, respectively (χ2 = 36.67, *p* < 0.001). Although the study results showed clear evidence of coordinated treatment, the attrition rate of study participants who did not complete the 7-month evaluation was over 60% (207 of 313). Additionally, the lack of a control group decreased validity and reliability for this study and its application. 

Based on the literature review focused on only vocational counseling or coordinated programs, study results indicated multiple common limitations, primarily a lack of randomization and control groups, as well as small sample sizes. Some studies did not report on the validity of instruments and treatment procedures. Additionally, the definition and/or composition of vocational counseling interventions varied depending on the geographical characteristics of the clinics, funding bodies, and backgrounds of service providers, while some studies adapted and used the same interventions from previous studies to explore the effectiveness of the interventions. Therefore, generalization of any of the study findings was limited, especially in providing vocational counseling interventions alone or through coordinated programs to multiple participant groups (i.e., gender, ethnicity, education level, substance use severity, and employability). 

## 5. Discussion and Suggestions for Practice and Research

The goal of vocational counseling interventions for consumers with SUDs and co-occurring psychiatric disorders is to help them have physically and mentally healthier independent lives. In order to achieve this goal, vocational rehabilitation counselors provide interventions focusing on vocational exploration, plan development, vocational assessment, and job placement. These interventions involve examining a variety of potential occupational goals and providing employment choices for consumers. As explored previously, the intervention should lead to changes in physical and psychological symptom severity, vocational skills, and employment outcomes. The evidence from 11 studies indicates that vocational counseling interventions are a likely fit for substance use treatment. Specifically, consumers who attend coordinated interventions do, overall, show fewer physical and psychological symptoms and more positive work outcomes after completing vocational programs compared to those receiving either substance use treatment or vocational counseling interventions alone [4,20]. 

A central theme across four types of intervention programs described in this review is consumer-centered recovery planning. Many consumers are unable to capitalize on employment opportunities and secure jobs after completing substance use treatment [4]. Researchers [20] emphasized that being vocationally underprepared is the primary reason that consumers have fewer employment opportunities. Although they completed substance use treatment successfully, they showed serious underemployment and unemployment rates. Insufficient job readiness and lack of vocational skills are critical barriers, while their SUDs are significantly decreased.

Moreover, according to the current research literature, vocational counseling and substance use treatment are conceptualized and tested separately. The current study builds on previous literature [11,12] indicating that these services should be inseparable. Similarly, employment rates have improved among those completing both programs successfully [11,40]. This study found increasing evidence that the results of two combined models (supported employment plus skills training and incorporated substance abuse treatment) show better employment outcomes than the other two models, which focused on only one intervention as a vocational service [37,38,40]. From this result, it can be reasoned that when receiving incorporated interventions, consumers might have higher self-efficacy expectations for possible treatment outcomes, such as reduced substance use symptoms and high employability, allowing them to maintain healthier and independent lifestyles. Because many consumers have histories of failure in the management of alcohol/drug use issues and their jobs, SUDs and vocational counseling interventions not only are effective in promoting alcohol/drug abstinence but also promote employability and job maintenance.

In addition, the evidence indicates that manual-based programs (e.g., *Helping Offenders Work*, *Work-it-out*, and *the Job Club Counselor’s Manual*) show better employment outcomes than other vocational counseling programs. In the collected data, the efficacy of non-manual programs is mixed, while manual-based programs show clear results in post-treatment and follow-ups. Butler and colleagues [33] found significant decreases in employment, drug, and family issue levels according to the ASI-5 in manual-based interventions. Two manual-based studies found that 55% and 70% of participants who completed the program were employed in competitive jobs, respectively [34,36].

This review found evidence for the effectiveness of vocational counseling and the necessity of incorporating interventions in SUDs and co-occurring fields. People who provide vocational counseling services have been educated and trained to provide recovery services to consumers [13]. These professionals are knowledgeable in the medical and psychosocial aspects of SUDs and life care management. These knowledge areas lead to holistic recovery planning for consumers to return to home, school, work, and their communities. Specifically, vocational rehabilitation counselors are well-suited and well-trained to improve consumers’ motivation and interest in employment [42]. The studies in this review provide a direct path to expand the roles of vocational rehabilitation counselors and the usability of incorporated interventions in SUDs and co-occurring fields. 

Although researchers’ suggestions on the usefulness of vocational counseling and integrative interventions in the field of SUD and co-occurrence continue to emerge, current services still focus on reducing alcohol and/or drug use or abstinence from alcohol [42,43]. One hopeful sign is that public awareness of drug addiction has increased, and the government has begun to take an interest in the recovery of drug addicts and their social participation and employment. Due to these social changes, cursory attention has been placed on multidimensional treatment approaches for SUDs, such as vocational counseling services to address employment-related concerns [44]. However, contrary to our buoyant hopes, recent research on integrative interventions with vocational counseling is still limited. Many researchers believe that for recovery from SUD to be sustainable, treatment options must shift to incorporate services beyond those solely for alcohol and/or drug use reduction or abstinence. For example, the application of vocational counseling as an ancillary treatment intervention after completing SUD treatment is supported by the significant role of employment in the disability and recovery process [25,45]. 

Incidences of SUDs are associated with employment. A greater percentage of unemployed adults 18 or older are classified with SUDs, even though the average educational level of individuals with SUDs is comparable to that of the general population [46]. Adamson and Sellman [47] suggested that because unemployment is predictive of a poor treatment outcome for SUDs, treatment may be improved by directly addressing unemployment. Based on a systematic review of the literature, the authors identified employment, as an indicator of social functioning, to be among the most consistent univariate predictors of treatment outcome. Furthermore, Melvin and colleagues [48] cited evidence that gainful employment is one of the strongest and most consistent predictors of post-treatment success and sobriety maintenance, with employed individuals more likely to engage in treatment, complete treatment, and remain substance-free after treatment.

Despite the apparent benefits of employment to consumers with SUDs and psychiatric disorders, few strategies that incorporate vocational counseling services into standard treatment have been developed. Kim [11] indicated that because many substance abuse clinicians are unfamiliar with both the theory and practical benefits of vocational counseling, the addition of vocational content to treatment has not occurred. Although various vocational interventions have been attempted in clinical settings, few empirically driven longitudinal studies have investigated vocational counseling services for populations with SUDs or psychiatric disorders [49]. Due to the paucity of studies, the development and integration of vocational interventions into SUD treatment has been slow. Moreover, while researchers have focused their efforts on exploring how to maximize the effects of standard treatment to reduce SUD and psychiatric issues, alternative interventions such as vocational counseling services have been neglected [50], and consumers’ post-treatment employment status has been disregarded or minimized as a treatment outcome measure [51].

Finally, the limitations in these studies are glaring: the clinical research standards applied to SUD treatment research must be applied for vocational counseling efficacy. In light of the ubiquity of evidence-based treatment as the standard for funding, vocational counseling research must rise to the use of randomized control groups with repeated measurements, and journals must embrace replication studies. National funding agencies must likewise provide resources for integrated vocational treatments by specialized professionals across research, training, and human services grant proposals.

## 6. Limitations

The review of vocational counseling interventions provided a unique opportunity to collect empirical evidence on consumers’ employment. In the review, several studies show positive employment results, such as improved employment rates, longer workdays, and higher earnings. In spite of these results, more evidence-based studies using RCT including two- and three-group comparisons are required in the field. The total number of studies was only 11, and most studies were classified as level I or level III. The next study should collect and analyze a larger number of studies in order to increase the generalization of the study results. Four types of vocational counseling programs were described, and multidisciplinary aspects provide many clear questions that might not be answered through this paper. This paper only shows that a program including vocational counseling interventions is able to be applied in substance use and co-occurring disorder fields. This review outlines evidence for vocational rehabilitation counseling interventions for people with SUDs and co-occurring psychiatric disorders, but it does not cover all mental health fields. In order to increase the usability of incorporated treatment, the American Counseling Association (ACA), Commission on Rehabilitation Counselor Certification (CRCC), Council for Accreditation of Counseling and Related Educational Programs (CACREP), and Council on Rehabilitation Education (CORE) would be required to emphasize the importance of vocational counseling and incorporated treatments in substance use treatment and the mental health field. Additionally, adequate manual-based training and a curriculum approved by multiple certification associations could be developed and evaluated.

## 7. Conclusions

In order to elevate employability, providing incorporated treatment to those undergoing SUD treatment would maximize positive effects toward reducing issue severity and increasing quality of life. However, researchers tend to focus primarily on substance use treatments that often reference employment as an outcome, without much attention to vocational counseling services’ usefulness in augmenting that aspect of recovery. Despite the sparse evidence, incorporated treatment that includes vocational counseling seems to be more effective. In order to expand the efficacy of incorporated treatment, more studies that explore synergistic effects are necessary.

## Figures and Tables

**Figure 1 ijerph-19-04674-f001:**
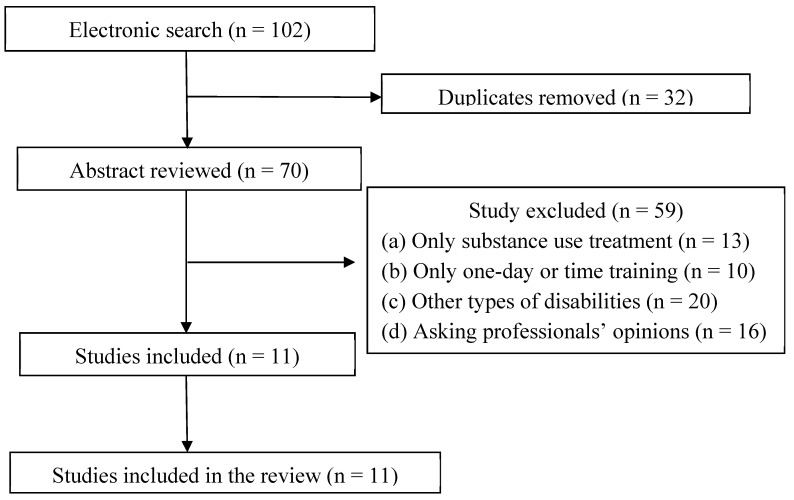
Flow chart of the phases of the search, exclusion, and inclusion process.

**Table 1 ijerph-19-04674-t001:** Level-of-evidence definitions.

Level I	Systematic reviews, meta-analyses, randomized controlled trials
Level II	Two groups, nonrandomized studies (e.g., cohort, case–control)
Level III	One group, nonrandomized (e.g., before and after, pretest and posttest)
Level IV	Descriptive studies that include analysis of outcomes (e.g., single-subject design)
Level V	Case reports and expert opinions, including consensus and narrative literature review)

From Sackett, Rosenberg, Muir Gray, Haynes, and Richardson (1996), p. 71.

**Table 2 ijerph-19-04674-t002:** Basic description of selected studies.

Category	Author (Year)	Levels of Evidence	*n*
Comprehensive work programs	Magura et al. (2004) [32]	A level I systematic review	-
	Butler et al. (2004) [33]	A level I RCT pre-posttest	194
	Kemp et al. (2004) [34]	A level III one-group study	245
Employment counseling and education programs	Lidz et al. (2004) [35]	A level I RCT	301
	Kidorf et al. (2004) [36]	A level III group intervention	254
Supported employment plus skills training	Staines et al. (2004) [37]	A level I RCT study	121
	Davis et al. (2012) [28]	A level I RCT study	75
	Kerrigan et al. (2004) [38]	A level III one-group study	80
Incorporated substance use treatment	Coviello et al. (2004) [39]	A level I RCT for a two-group study	109
	McLellan et al. (2003) [40]	A level III study	529
	Kim (2013) [11]	A level III one-group study	313

## Data Availability

The authors confirm that the data supporting the findings of this study are available within the article.

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
