# Peer review of "The Effect of Vocational Counseling Interventions for Adults with Substance Use Disorders: A Narrative Review"

_ijerph, 2022, doi:10.3390/ijerph19084674_

Round 1

Reviewer 1 Report

This review in- 9 cluded 19 articles related to vocational counseling interventions, which are either incorporated with 10 substance use treatment or not. The results of this review revealed strong efficacy for vocational 11 counseling services to result in part-time and full-time jobs. Study designs had some limitations, 12 and few studies employed randomized control trials. Overall, it is a comprehensive review and merits to be published in International Journal of Environmental Research and Public Health. However, major revision should be done to further improve the quality of this manuscript.

  1. The future development of this area is suggested to have more discussions.
  2. The author should put forward his own opinions in this review rather than describe the research of others.
  3. The author should provide a schematic diagram to express the theme and views of this review.
  4. There are many problems in the references.

Author Response

Point 1. The future development of this area is suggested to have more discussions.

- Response 1: Added 3 paragraphs to keep up with the latest trends in the DISCUSSION section

Point 2. The author should put forward his own opinions in this review rather than describe the research of others.

- Response 2: Added author’s opinions in the DISCUSSION section

Point 3. The author should provide a schematic diagram to express the theme and views of this review.

- Response 3: Added one table (Basic Description of aa studies) and revised the flow chart

Point 4. There are many problems in the references.

- Response 4: Changed reference formats

Reviewer 2 Report

The paper ‘The Effect of Vocational Counseling Interventions for Adults with Substance Use Disorders: A Narrative Review’ by Kim et al. investigated the evidence for vocational counseling interventions’ impact on SUD services and co-occurring disorders. The work quality is good enough to be considered in Int. J. Environ. Res. Public Health. However, some format corrections should be done before publication.

Author Response

The paper ‘The Effect of Vocational Counseling Interventions for Adults with Substance Use Disorders: A Narrative Review’ by Kim et al. investigated the evidence for vocational counseling interventions’ impact on SUD services and co-occurring disorders. The work quality is good enough to be considered in Int. J. Environ. Res. Public Health. However, some format corrections should be done before publication.

- Changed reference formats

Reviewer 3 Report

The manuscript presented for the review is very interesting and it analyses literature referring to effect of vocational counselling interventions on employment of consumers with substance use disorders. The Authors indicated a strong relationship of vocational counselling services with both part-time and full-time jobs. They also found a limitations of presented studies and pointed the need of future work that should be established.

The manuscript is suitable for the publication in International Journal of Environmental Research and Public Health after minor revision. There are some issues that should be revised by the Authors.

According to Instructions for Authors reference numbers should be placed in the text in square brackets and placed before the punctuation and not as names with a date.

The References section is also incompatible with the Instructions for Authors. The order of names and surnames is inverted in every second and next Author. The initials of names should be used. The names of Journals should be abbreviated. The year stands in incorrect place. Punctuation should be corrected. All references should be written according to the Instructions for Authors.

The biggest concern about the manuscript is the timeliness of the analysed literature. The Authors have searched for publications from January 1990 to January 2021, which gives a 31 years time interval. According to study selection criteria, the Authors have chosen 19 articles for the analyses. Most of analysed articles are from 2004 (8 articles) which is 18 years ago. Nine articles are older and only 2 of them are from 2012 and 2013. I can understand that probably it is due to the selection criteria but it also makes the question if the data is up-to-date. The economic situation of a given country, people's wealth, educational programs, career counselling programs may change over time, which can significantly affect the employment of people with substance use disorders. Drug addiction and addiction treatment programs may also change. I would suggest to add a short paragraph explaining the lack of newer sources and mentioning the up-to-date situation even outside the specified criteria.

For example in line 78, there is a sentence which state that a “new models have been suggested…”, but the reference is from 2010.

Author Response

- Author's notes to the reviewer

The manuscript presented for the review is very interesting and it analyses literature referring to effect of vocational counselling interventions on employment of consumers with substance use disorders. The Authors indicated a strong relationship of vocational counselling services with both part-time and full-time jobs. They also found a limitations of presented studies and pointed the need of future work that should be established.

The manuscript is suitable for the publication in International Journal of Environmental Research and Public Health after minor revision. There are some issues that should be revised by the Authors.

According to Instructions for Authors reference numbers should be placed in the text in square brackets and placed before the punctuation and not as names with a date.

- Changed reference formats

The References section is also incompatible with the Instructions for Authors. The order of names and surnames is inverted in every second and next Author. The initials of names should be used. The names of Journals should be abbreviated. The year stands in incorrect place. Punctuation should be corrected. All references should be written according to the Instructions for Authors.

- Changed reference formats

The biggest concern about the manuscript is the timeliness of the analysed literature. The Authors have searched for publications from January 1990 to January 2021, which gives a 31 years time interval. According to study selection criteria, the Authors have chosen 19 articles for the analyses. Most of analysed articles are from 2004 (8 articles) which is 18 years ago. Nine articles are older and only 2 of them are from 2012 and 2013. I can understand that probably it is due to the selection criteria but it also makes the question if the data is up-to-date. The economic situation of a given country, people's wealth, educational programs, career counselling programs may change over time, which can significantly affect the employment of people with substance use disorders. Drug addiction and addiction treatment programs may also change. I would suggest to add a short paragraph explaining the lack of newer sources and mentioning the up-to-date situation even outside the specified criteria.

- Changed selection criteria based on the reviewer's recommendation and finally the number of articles reviewed are changed from 19 to 11. 

- Added 3 paragraphs to keep up with the latest trends and ㅅhe limitations of the number of studies were emphasized again in the LIMITATIONS section.

For example in line 78, there is a sentence which state that a “new models have been suggested…”, but the reference is from 2010.

- Added 6 latest references

Round 2

Reviewer 1 Report

All issues have been addressed and so I recommend its publication in IJERPH.